# Local computational methods to improve the interpretability and analysis of cryo-EM maps

Satinder Kaur[1,5], Josue Gomez-Blanco[1,5], Ahmad A. Z. Khalifa[1], Swathi Adinarayanan[1], Ruben Sanchez-Garcia [2], Daniel Wrapp [3], Jason S. McLellan [3], Khanh Huy Bui [1] & Javier Vargas [4✉]

Cryo-electron microscopy (cryo-EM) maps usually show heterogeneous distributions of *B*-factors and electron density occupancies and are typically *B*-factor sharpened to improve their contrast and interpretability at high-resolutions. However, 'over-sharpening' due to the application of a single global *B*-factor can distort processed maps causing connected densities to appear broken and disconnected. This issue limits the interpretability of cryo-EM maps, i.e. ab initio modelling. In this work, we propose 1) approaches to enhance high-resolution features of cryo-EM maps, while preventing map distortions and 2) methods to obtain local *B*-factors and electron density occupancy maps. These algorithms have as common link the use of the spiral phase transformation and are called LocSpiral, LocBSharpen, LocBFactor and LocOccupancy. Our results, which include improved maps of recent SARS-CoV-2 structures, show that our methods can improve the interpretability and analysis of obtained reconstructions.

[1] Departament of Anatomy and Cell Biology, McGill University 3640 Rue University, Montréal, QC, Canada. [2] Biocomputing Unit, Centro Nacional de Biotecnología-CSIC C/Darwin 3, Cantoblanco, Madrid, Spain. [3] Department of Molecular Biosciences, The University of Texas at Austin, Austin, TX, USA. [4] Departamento de Óptica, Universidad Complutense de Madrid, Madrid, Spain. [5] These authors contributed equally: Satinder Kaur, Josue Gomez-Blanco. ✉email: jvargas@fis.ucm.es

Cryo-electron microscopy (cryo-EM) has become a mainstream technique for structure determination of macromolecular complexes at close-to-atomic resolution and ultimately for building an atomic model[1,2]. With its unique ability to reconstruct multiple conformations and compositions of the macromolecular complexes, cryo-EM allows the understanding of the structural and assembly dynamics of macromolecular complexes in their native conditions[3–5]. However, the presence of heterogeneity in cryo-EM maps leads to high variability in resolution within different regions of the same map. This directs to challenges and errors in the process of building an atomic model from a cryo-EM reconstruction. Additionally, current reconstructions from cryo-EM do not provide essential information to build accurate ab initio atomic models as atomic Debye–Waller factors (B-factors) or atomic occupancies, while their counterparts from X-ray crystallography do by analysing the attenuation of scattered intensity represented at Bragg peaks.

Cryo-EM structures exhibit loss of contrast at high-resolution coming from many different sources, including molecular motions, heterogeneity and/or signal damping by the transfer function of the electron microscope (CTF). Interpretation of high-resolution features in cryo-EM maps is essential to understanding the biological functions of macromolecules. Thus, approaches to compensate for this contrast loss and improve map visibility at high-resolution are crucial. This process is usually referred to as 'sharpening' and is typically performed by imposing a uniform B-factor to the cryo-EM map that boosts the map signal amplitudes within a defined resolution range. When the map is sharpened with increasing positive B-factors, the clarity and map details initially improve, but eventually, the map becomes worse as the connectivity is lost, and the map densities appear broken and noisy. In the global sharpening approach[6–8], the B-factor is automatically computed by determining the line that best fits the decay of the spherically averaged noise-weighted amplitude structure factors, within a resolution range given by [15–10 Å, $R_{max}$], with $R_{max}$ the maximum resolution in the map given by the Fourier Shell Correlation (FSC). More recently, the AutoSharpen method within Phenix[9] calculates a single B-factor that maximises both map connectivity and details of the resulting sharpened map. AutoSharpen automatically chooses the B-factor that leads to the highest level of detail in the map, while maintaining connectivity. This combination is optimised by maximising the surface area of the contours in the sharpened map.

The approaches presented above are global, so the same signal amplitude scaling is applied to map regions that may exhibit very different signal to noise ratios (SNRs) at medium/high-resolutions. Thus, cryo-EM maps showing inhomogeneous SNRs (and resolutions) can result in sharpened maps that show both over-sharpened and under-sharpened regions. The former may be strongly affected by noise and broken densities, while the latter may present reduced contrast at high-resolutions. Both cases make it difficult or even impossible to interpret the biological relevance of these regions or even the whole map[10]. Thus, local sharpening methods have been proposed to overcome these limitations[11,12]. LocScale approach[11] compares radial averages of structure factor amplitudes inside moving windows between the experimental and the atomic density maps. After, the method modifies locally the map amplitudes of the experimental map in Fourier space to rescale them accordingly to those of the atomic map. This approach requires as input a complete atomic model (without major gaps) fitted to the cryo-EM map to be sharpened, which is not always available. In addition, the size of the moving window should be provided and depending on the quality of the map to be sharpened, this process may lead to overfitting. More recently, the LocalDeblur method[12] proposed an approach for map local sharpening using as input an estimation of the local

resolution. The method assumes that the map local density values have been obtained by the convolution between a local isotropic low-pass filter and the actual map. This local low-pass filter is assumed Gaussian-shaped so that the frequency cutoff is given by the local resolution estimation.

In X-ray crystallography, the B-factor (also called temperature value or Debye–Waller factor) describes the degree to which the electron density is spread out, indicating the true static or dynamic mobility of an atom and/or the positions where errors may exist in the model building. The B-factor is given by $B_i = 8\pi^2 u_i^2$, where $u_i^2$ is the mean square displacement for atom $i$. These atomic B-factors can be experimentally measured in X-ray crystallography, introduced as an amendment factor of the structure factor calculations since the scattering effect of X-ray is reduced on the oscillating atoms compared to the atoms at rest[13]. B-factors can be further refined by model building packages, i.e. Phenix[14] or Refmac[15] to improve the quality and accuracy of atomics models. Although B-factors are essential to 'sharpen' cryo-EM maps at high-resolution, they also provide key information to analyse cryo-EM reconstructions. Effective B-factors are used to model the combined effects of issues such as molecular drifting due to charging effects, macromolecular flexibility or possible errors in the reconstruction workflow that lead to a signal fall-off[6,16,17]. However, cryo-EM maps are usually analysed with a single B-factor, even though maps may largely differ in different regions. Thus, methods to determine local B-factors are much needed to accurately analyse cryo-EM maps and improve the quality of fitted atomic models. Another local parameter usually provided by X-ray crystallography in contrast with cryo-EM are atomic occupancies (or Q-values). The occupancy estimates the presence of an atom at its mean position and it ranges between 0.0 to 1.0. Note that these parameters can be also refined by model building packages if the electron density map is of sufficient resolution. To our knowledge, currently, there is not any available method to estimate local occupancies from cryo-EM maps, even though this information (in addition to local B-factors) is essential to building accurate atomic models. For example, in ref. [18] authors found that 31% of all models examined in this analysis possess unrealistic occupancies or/and B-factor values, such as all being set to zero or other unlikely values. They also reported that 40% of models analysed show cross-correlations between cryo-EM maps and respective models below 0.5, and they indicated as a possible hypothesis an incomplete optimisation of the model parameters (coordinates, occupancies and B-factors).

In this work, we propose semi-automated methods to enhance high-resolution map features to improve their visibility and interpretability. More importantly, these approaches do not require input parameters as fitted atomic models or local resolution maps, which reduces the possibility of overfitting. In particular, our proposed local map enhancement approach (LocSpiral) is robust to maps affected by inhomogeneous local resolutions/SNRs, thus the method strongly improves the interpretability of these maps. Secondly, we also propose approaches to determine local B-factors and density occupancy maps to improve the analysis of cryo-EM reconstructions. The link between the different proposed approaches is the use of the spiral phase transform to estimate a modulation or amplitude map of the cryo-EM reconstruction at different resolutions.

## Results

In this section, we first provide a brief and comprehensive description of the approaches developed in this work. A deeper and more technical explanation of these methods is given in the 'Methods' section at the end of the manuscript. Then, we present

results obtained by our approaches in a variety of situations. We tested our proposed methods with five different samples ranging from near-atomic single-particle reconstructions (~1.54 Å) to maps with more modest resolutions (~6.5 Å). In all cases, we compared our results with the ones provided by the Relion postprocessing approach[7,19].

**Overview of the proposed methods**. The input parameters of the different methods (LocSpiral, LocBSharpen, LocBFactor and LocOccupancy) is the unfiltered map to process, a resolution range given by $[R_{min}, R_{max}]$ and, in some cases, a tight solvent mask. The different algorithms start by filtering the input map to a given resolution $1/\omega$ within the resolution range. Then, the 3D spiral phase transform is calculated to factorise in real space the filtered map into amplitude and a phase map as

$$V_\omega(\mathbf{r}) = m_\omega(\mathbf{r})\cos(\varphi_\omega(\mathbf{r})) \quad (1)$$

The amplitude map $m_\omega(\mathbf{r})$ is related to the 'strength' of the local map signal at resolution $1/\omega$, while the phase map refers to its shape and it is limited to the $[-1, +1]$ range. The different methods proposed here are based on the analysis of the amplitude maps. In some cases, the approaches compute new amplitude maps $(\breve{m}_\omega(\mathbf{r}))$, which are used to determine a sharpened map (LocSpiral, LocBSharpen) as

$$\breve{V}(\mathbf{r}) = \sum_\omega \breve{V}_\omega(\mathbf{r}) = \sum_\omega C_{ref,\omega}(\mathbf{r})\breve{m}_\omega(\mathbf{r})\cos(\varphi_\omega(\mathbf{r})) \quad (2)$$

with $C_{ref,\omega}(\mathbf{r})$ a SNR weighting parameter (please see 'Methods' section). In other cases, the amplitude maps are further analysed to provide local $B$-factor maps (LocBFactor) or a local occupancy map estimation (LocOccupancy).

In LocSpiral at every resolution inside the resolution range, the amplitude map is compared locally with a noise threshold value computed from the 90–95% quantile of the empirical noise/background distribution at this resolution. This empirical distribution is generated collecting the amplitude values at resolution $1/\omega$ for all voxels outside the tight solvent mask. The computed noise threshold is then used to obtain a new normalised and filtered amplitude map, $\breve{m}_\omega(\mathbf{r})$, which is used to reconstruct the sharpened map as shown in Eq. (2).

In LocBSharpen the amplitude map at a resolution $1/\omega_0$ ($m_{\omega_0}(\mathbf{r})$) is stored. The resolution $1/\omega_0$ is provided by the user and is typically 15–10 Å. In the process of building the sharpened map, the new amplitude map $\breve{m}(\mathbf{r})$ at any resolution equal or higher than $1/\omega_0$ is equal to $m_{\omega_0}(\mathbf{r})$, while for the rest of resolutions inside the resolution range, $\breve{m}_\omega(\mathbf{r})$ is equal to $m_\omega(\mathbf{r})$.

In LocBFactor the amplitude maps at different resolutions inside the defined resolution range are used to estimate map local $B$-factors. A typical resolution range is of $[15, R_{max}]$ Å, being $R_{max}$ the global map resolution. To compute the local $B$-factors, the method obtains at every voxel $\mathbf{r}$ the linear fitting between $\log(C_{ref,\omega}(\mathbf{r})m_\omega(\mathbf{r}))$ and $\omega^2$ within the resolution rage. The method provides as output the $B$-map (local $B$-factor map) and the A-map (local values of the logarithm of structure factor amplitudes at 15 Å).

In LocOccupancy, the local occupancy map is estimated comparing the amplitude map with a macromolecule density threshold for every resolution inside the defined resolution range. The macromolecule density threshold at a given resolution indicates the density value at which we are confident that the electron density occupancy is of 100% at this resolution. This threshold is obtained from the empirical macromolecule amplitude probability distribution $(m_\omega^M)$ at frequency $\omega$. This amplitude probability distribution is calculated from density

values at voxels that are included inside the solvent mask. From this distribution, the macromolecule density threshold may be calculated from the macromolecule amplitude value corresponding to the 25% quantile, given by $m_\omega^M(q = 25\%)$. Then, for every voxel and resolution within the resolution range, the amplitude map $m_\omega(\mathbf{r})$ is compared with $m_\omega^M(q = 25\%)$, providing a value between 0 and 1. Finally, the average value over all resolutions is computed and provided as an estimation of the map occupancy within the resolution range.

**Polycystin-2 (PC2) TRP channel**. First, we analysed a single-particle reconstruction of the polycystin-2 (PC2) TRP channel (EMDataBank: EMD-10418)[20]. In this case, we focussed on showing the capacity of LocSpiral approach, though, for the sake of consistency, we also show results of obtained $B$-factor and occupancy maps. The original publication reports a resolution of 2.96 Å with a final $B$-factor to be used for sharpening of −84.56 Å² (slope of Guinier plot fitting equal to −21.14 Å²).

In Fig. 1A, we show maps with high threshold values obtained by LocSpiral and by the postprocessing method of Relion 3[7,19]. The map densities are similar in the inner core of the protein as can be seen from the solid red rectangle in the figure, where we show a zoomed view of LocSpiral and Relion maps of the region indicated in the red rectangles over the maps. However, the map densities are quite different in the outer regions, where the Relion map shows thin and broken densities. In addition, we show comparisons of fitted densities with the corresponding atomic model (PDB ID: 6t9n) of two α-helices and one loop. The asterisks label results obtained by LocSpiral. The residues marked with a red arrow were used to adjust the threshold values between maps. These comparisons show that the map obtained by LocSpiral shows fewer fragmented and broken densities and better coverage of the atomic model, helping in the interpretation of the maps and in the process of building accurate atomic models. In Supplementary Fig. 1, we show additional figures comparing LocSpiral and Relion postprocessing maps.

We also compared the performance of LocSpiral with other methods, including LocalDeblur, our proposed local $B$-factor correction method (LocBSharpen) and the global $B$-factor correction approach as implemented in Relion. The results are shown in Supplementary Note 1, Supplementary Table 1 and Supplementary Fig. 2 where we also provide results obtained by LocBFactor and LocOccupancy methods.

**Pre-catalytic spliceosome**. Next, we processed the Saccharomyces cerevisiae pre-catalytic B complex spliceosomal single particles deposited in EMPIAR (EMPIAR 10180)[4,21]. This dataset exhibits a high degree of conformational heterogeneity, thus, it represents a perfect use case to test our proposed approaches. We used the approach described in ref. [22] to obtain a reconstruction at 4.28 Å resolution after Relion postprocessing[7,19]. In the 'Methods' section, we provide a detailed description of the image processing workflow used to obtain this reconstruction. The unfiltered map provided by Relion autorefine was used as input to LocSpiral, LocBFactor and LocOccupancy.

We first show results obtained by LocSpiral method for this highly heterogeneous case. In Fig. 1B, we show maps at different orientations and similar threshold values obtained by LocSpiral and by the postprocessing method of Relion 3[7,19]. As before, the LocSpiral map shows fewer fragmented and broken densities, especially in the flexible part of the spliceosome reconstruction, and enhanced details in the central core portion improving the visibility of the reconstruction.

We then concentrate on showing the capacity of LocBFactor method. In Fig. 2A, we show a central slice along the $Z$ axis of this

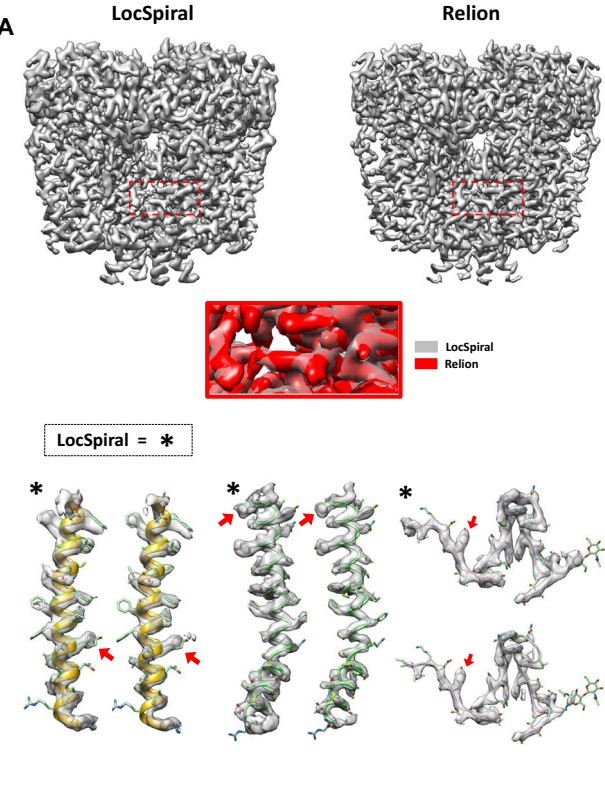

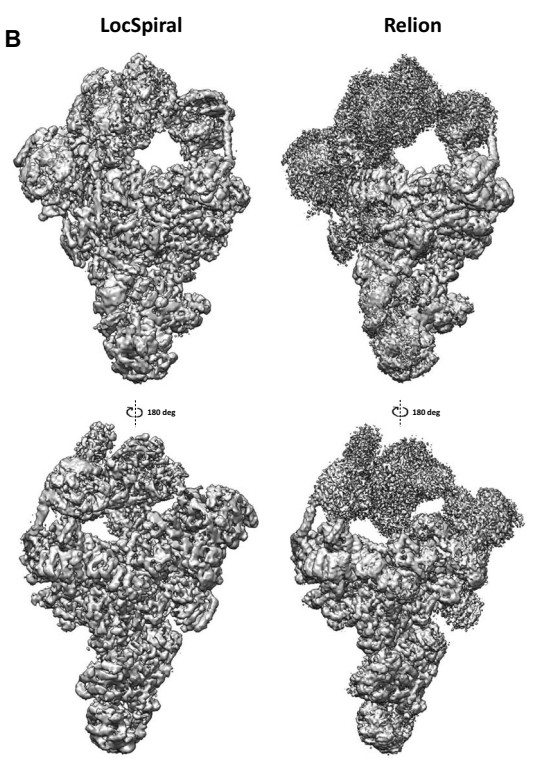

**Fig. 1 Capacity of LocSpiral to improve the interpretability of cryo-EM maps. A** Top: sharpened maps of the TRP channel obtained by LocSpiral (left) and Relion postprocessing (right) methods. The threshold values are adjusted to provide similar densities in the core inner part of the protein. The red square in the figure shows a zoomed view of the protein inner core where both maps (LocSpiral and Relion) are superimposed. Relion map appears in red colour, while LocSpiral is in grey. Bottom: Fitted map densities (LocSpiral and Relion) with the corresponding atomic model (PDB ID: 6t9n) of two α-helices and one loop. The asterisks mark results obtained by LocSpiral approach. The residues marked with a red arrow were used to adjust the threshold values between maps. **B** Spliceosome maps at different orientations and similar threshold values obtained by LocSpiral and the postprocessing method of Relion.

map with several points marked with coloured squares. These points show parts of the map that correspond to clear spliceosome densities (green and red), flexible and low-resolution spliceosomal regions (yellow and blue) and background (magenta). Figure 2B shows the corresponding Guinier plots at these locations. Solid lines represent measured values of the logarithm of SNR-weighted structure factor amplitudes, while

dashed lines show fitted curves. This figure also provides the obtained $B$-factors for the different curves. The Guinier plots and $B$-factors are determined within a resolution range of 15 Å to the FSC resolution, given by 4.28 Å. As can be seen from Fig. 2B, the red and green curves, which correspond to clear spliceosomal densities, present high amplitude values at 15 Å, while the yellow, blue and magenta curves show low amplitudes at 15 Å and a flat profile within the resolution range. In Fig. 2B, we also show in the black curve, the Guinier plot of the noise/background amplitudes obtained from the 90–95% quantile of the empirical noise/ background distribution for reference. The discontinuous black line indicates the linear fit of this noise Guinier plot. Comparing the yellow, blue, magenta, and black curves, it is clear that these plots are below our noise level and that the shape of these curves is similar to that of the noise curve. Thus, these $B$-factors describe mainly noise $B$-factors that show how the noise signal fall off inside the used resolution range and they should be filtered out from our $B$-factor map. Moreover, Fig. 2C shows the spliceosome map coloured according to the occupancy map obtained by LocOccupancy using a resolution range of [30, 10] Å. From Fig. 2C, we see that the flexible and moving parts of the spliceosome, like the ones indicated with the yellow and blue points in Fig. 2A, show low occupancies (close to zero) within the used resolution range. Figure 2D renders the spliceosome map coloured with the obtained $B$-factor map to be used for sharpening (slope of the local Guinier plot multiplied by 4). In Fig. 2D the noise $B$-factors ($B$-factors obtained from amplitudes below the noise level for the used resolution range) are filtered out and appear with black colour. Note that Guinier plots at regions with amplitudes below the noise level are dominated by the noise signal and describe the noise signal fall-off inside the used resolution range. The noise signal presents typically a flat spectrum, thus, artefactual close to zero $B$-factors, which are not in agreement with the concept of $B$-factor as a measure of position uncertainty or disorder. Figure 2E shows the corresponding local resolution map as obtained by Resmap[23] of the spliceosome reconstruction. As can be seen from this figure, the local resolution values of the flexible parts (helicase and SF3b domains) are lower than the others and within a range of [10, 15] Å. Consequently, the obtained amplitudes for these flexible parts within the resolution range of [15, 4.28] Å are dominated by the noise/background signal. The average inside a solvent mask of the signal $B$-factors ($B$-factors obtained from amplitude values above the noise level for the used resolution range) is −567.62 Å², while the value reported by Relion postprocessing is −158.08 Å². Note that Relion postprocessing does not filter out regions dominated by noise/background when computing the global $B$-factor. As mentioned before, regions dominated by the noise signal within the used resolution range present artefactual low $B$-factors. Consequently, this global $B$-factor may be overestimated. A more

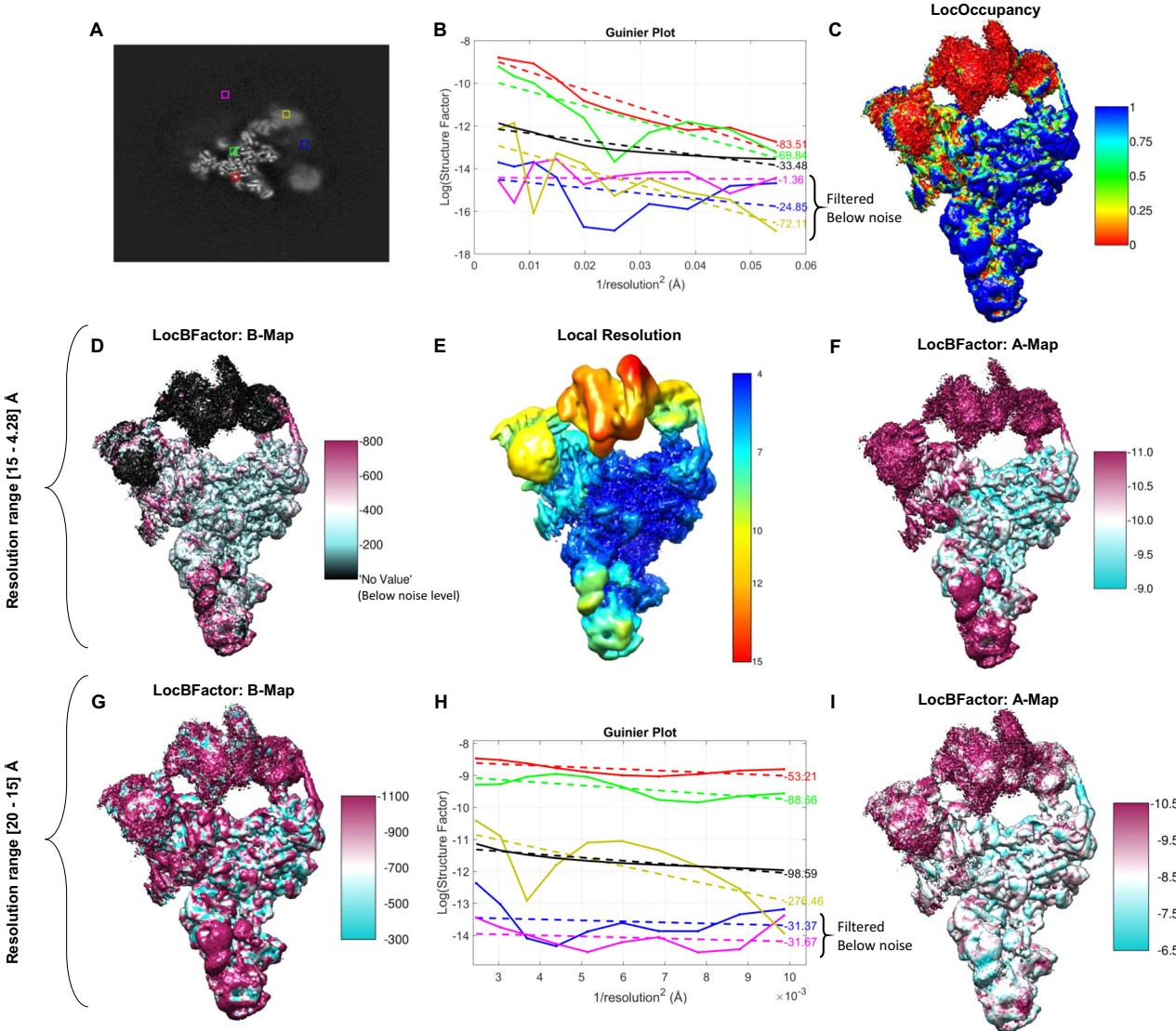

**Fig. 2 Results obtained by LocBFactor and LocOccupancy for the Saccharomyces cerevisiae pre-catalytic B complex spliceosome sample. A** Central slice along $Z$ axis of the obtained unsharpened map using EMPIAR 10180 single particles. Coloured squares mark parts of the map corresponding to clear spliceosome densities (green and red), flexible and low-resolution spliceosomal regions (yellow and blue) and background (magenta). **B** Guinier plots at map points indicated in the coloured squares using a resolution range of [15–4.28] Å. Solid lines represent SNR-weighted values of the logarithm of structure factor amplitudes, while discontinued lines show the fitted lines. **C** Spliceosome map coloured with the obtained occupancy map by LocOccupancy. The occupancy ranges from 0 (red colour) to 1 (blue colour), indicating no macromolecular density and full occupancy, respectively. **D** Spliceosome map coloured with the $B$-factor map to be used for sharpening (slope of the local Guinier plot multiplied by 4) obtained by LocBFactor using a resolution range of [15–4.28] Å. The local $B$-factor values in the figure range approximately between −800 and −200 Å². In this figure, noise $B$-factors ($B$-factors obtained from amplitudes below the noise level for the used resolution range) are filtered out and appear with black colour. **E** Local resolution map obtained by Resmap approach. The local resolution ranges between 4 (blue colour) and 15 Å (red colour). **F** Spliceosome map coloured with the obtained A map (local values of the logarithm of structure factor amplitudes at 15 Å). The values range between −11.0 (magenta colour) and −9.0 (cyan colour) approximately. **G** Spliceosome map coloured with the $B$-factor map ($B$-map) obtained by LocBFactor using a resolution range of [20–15] Å. The $B$-factor values range between −1100 (magenta colour) and −300 (cyan colour) approximately. **H** Guinier plots at map points indicated in the coloured squares using a resolution range of [20–15] Å. Solid lines represent SNR-weighted values of the logarithm of structure factor amplitudes, while discontinued lines show the fitted lines. **I** Spliceosome map coloured with the obtained A map (local values of the logarithm of structure factor amplitudes at 20 Å). The values range between −10.5 (magenta colour) and −6.5 (cyan colour) approximately.

detailed description of this point is given in Supplementary Note 3: $B$-factor analysis of low and high-resolution maps. In Fig. 2F, we show the local values of the logarithm of the structure factor's amplitudes at 15 Å (A map). As expected, this map shows low amplitudes at highly flexible and moving regions. We have recalculated $B$-factors using a new resolution range of [20, 10] Å. The results are shown in Fig. 2G–I. As can be seen from these figures, now the flexible parts show unfiltered low signal $B$-factors

and low amplitudes at 20 Å. However, it is important to note that at this resolution range, the $B$-factors are dominated by the molecular shape and solvent contrast and not by resolution limiting factors such as errors in the reconstruction procedure (as the presence of heterogeneity), radiation damage or imaging imperfections, for example[6]. Consequently, it is not recommended to use a resolution range of [20, 10] Å as obtained $B$-factors may not be used to evaluate map quality.

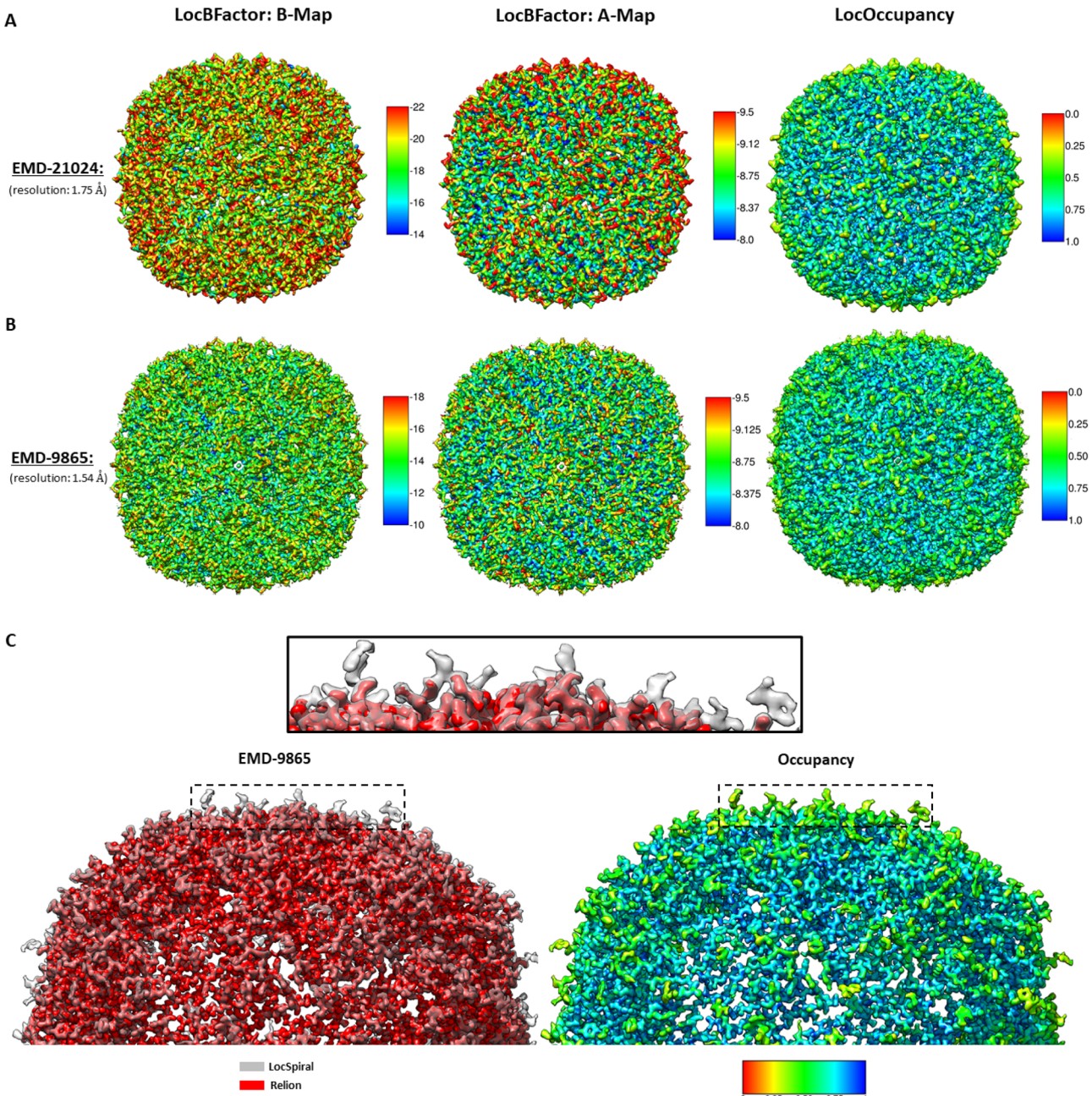

**Fig. 3 Results obtained by LocBFactor, LocOccupancy and LocSpiral for apoferritin sample.** Obtained *B*-maps (local *B*-factor map corresponding to the slope of the local Guinier plots), A-maps (local values of the logarithm of structure factor amplitudes at 15 Å) and occupancy maps by LocBFactor and LocOccupancy for EMD-21024 (**A**) and EMD-9865 (**B**). The *B*-factor ranges between [−22, −14] Å² in **A** and [−18, −10] Å² in **B**. The A-map ranges between [−9.5, −8.0] in **A** and **B**. The occupancy ranges between [0, 1] in **A** and **B**. In **C**, we show on the left side, superimposed sharpened maps obtained by LocSpiral (grey colour) and Relion (red colour) for EMD-9865. The black rectangle shows a zoomed view of the region indicated with the dashed rectangles. On the right, we show the respective occupancy map obtained by LocOccupancy at the same orientation that these sharpened maps. In this figure, 0 (red colour) indicates no density occupancy and 1 (blue colour) full occupancy.

**Apoferritin**. We have also applied these techniques to recently reported high-resolution cryo-EM reconstructions of mouse apoferritin: EMD-9865 and EMD-21024. The reported global resolution of these reconstructions is 1.54 and 1.75 Å for EMD-9865 and EMD-21024, respectively.

In Fig. 3A, B, we show the results obtained by LocBFactor (B and A maps) and LocOccupancy methods (occupancy maps). The resolution range used to estimate the B and A maps was between 15 Å to the reported global resolution for both cases. The occupancy maps were calculated for these high-resolution maps

between 5 Å to the global resolution. As can be seen from Fig. 3B, EMD-9865 shows lower *B*-factors and higher local amplitudes than EMD-21024, indicating a better-quality reconstruction, however, the low values of both B maps indicate the high quality of these reconstructions. In both cases, the highest *B*-factors are in the outer regions of the protein. Moreover, local occupancies show similar maps for both cases, showing occupancies as low as approximately 0.5 at the outer part and indicating the presence of flexibility in these outer residues. Note that the obtained average and standard deviation of *B*-factors inside a solvent mask is of

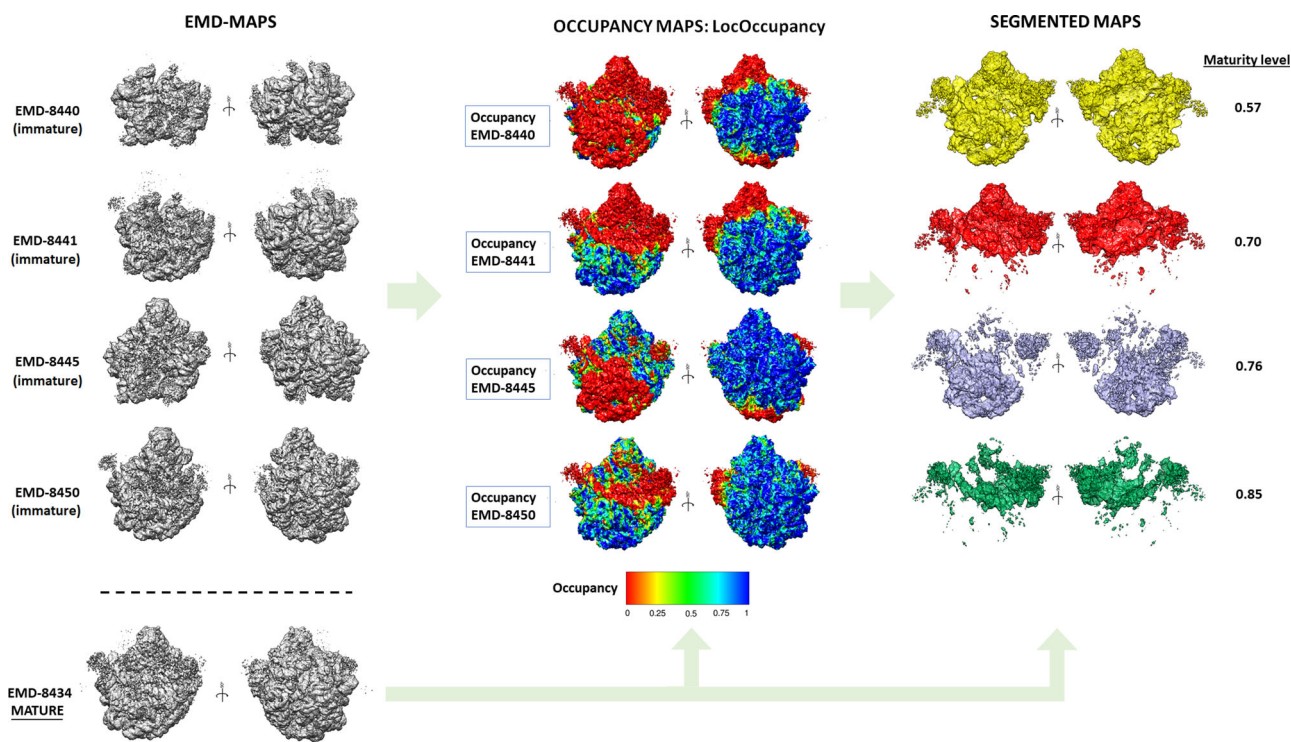

**Fig. 4 Results obtained by LocOccupancy for immature 50S ribosomes.** First column: immature maps at different orientations as deposited in EMDB. Second column: obtained occupancy maps by LocOccupancy, where the mature 50S ribosome (EMDB-8434) is coloured with corresponding occupancy maps obtained from the immature ribosomes. The occupancy ranges between [0, 1]. Third column: Segmented maps showing the densities that are missing in the different immature maps when compared to the mature 50S reconstruction and obtained maturity levels. The different colours (yellow, red, purple, green) label the different corresponding segmented regions for each case.

−56 and 7.20 Å$^2$ (EMD-9865) and −78 and 8.93 Å$^2$ (EMD-21024), respectively, which reflects the high quality of these reconstructions.

In Fig. 3C, we show sharpened maps obtained from EMD-9865 by LocSpiral and by the postprocessing method of Relion 3. The LocSpiral map is shown in grey colour, while the Relion map is rendered in red. The solid black rectangle shows a zoomed view of the outer region of the protein, which is indicated with the dashed black rectangles in the figure. Supplementary Fig. 3 shows that the extra densities that appear in the LocSpiral map correspond to missing residues in EMD-9865. Additional, at the right of Fig. 3C, we show the respective occupancy map obtained by LocOccupancy at the same orientation that these sharpened maps. As can be seen from Fig. 3C, the LocSpiral map shows fewer fragmented and broken densities, especially in the parts of the map that shows low occupancies. We compute also EMRINGER and MolProbity scores[24] between these maps (EMD-9865 and LocSpiral) and the atomic model (PDB 6v21) after refining the structure against corresponding maps by Phenix real_space_refine approach[25] using 5 refining iterations. The results obtained are shown in Supplementary Table S1.

**Immature prokaryote ribosomes.** We processed immature ribosomal maps of the bacterial large subunit[3]. These maps were obtained after depletion of bL17 ribosomal protein and are publicly available from the Electron Microscopy Data Bank (EMDB) (EMD-8440, EMD-8441, EMD-8445, EMD-8450, EMD-8434)[26]. In this case, we focussed on showing the capacity of LocOccupancy to interpret and analyse reconstructions showing a high degree of compositional heterogeneity.

Figure 4 shows the obtained results. The first row shows the different maps to be processed as deposited in the EMDB. Next, we show the obtained occupancy maps by LocOccupancy, where

the mature 50S ribosome (EMD-8434) is coloured according to corresponding occupancy maps. The resolution range used was [30, 10] Å. These figures clearly show regions that are lacking in the different immature maps with respect to the mature map. Thus, occupancy maps were used to create binary masks to segment the mature 50S ribosome map, extracting after the densities that are missing in the respective immature maps. These densities are shown in the third column of Fig. 4 with different colours (yellow, red, indigo and green). The obtained occupancy maps also allow us to define a 'maturity level' index. This index is calculated by comparing the number of voxels activated in the solvent mask of the mature 50S reconstruction with the ones in the occupancy masks (see methods section for a more detailed description). As can be seen from Fig. 4, the larger the unfolded regions in the immature maps are, the smaller the maturity level is. This maturity level index allows us to quantitatively sort the different immature maps in a spectrum according to their maturity.

In the Supplementary Note 2 and Supplementary Fig. 4, we further show the advantages of LocSpiral and LocBFactor approaches in these highly heterogeneous datasets compared to the global sharpening approach.

**SARS-CoV-2.** We have processed recent cryo-EM maps of the SARS-CoV-2 spike (S) glycoprotein[27,28]. These maps include cryo-EM reconstructions of the SARS-CoV-2 spike in the pre-fusion conformation with a single receptor-binding domain (RBD) up (EMD-21375) and after imposing C3 symmetry in the refinement to improve visualisation of the symmetric S2 subunit (EMD-21374). We also processed additional cryo-EM reconstructions from the Veesler lab of the SARS-CoV-2 spike glycoprotein with three RBDs down (EMD-21452) and the SARS-CoV-2 spike ectodomain structure (EMD-21457) with a single

RBD up. The reported global resolution of these maps is 3.46, 3.17, 2.8 and 3.2 Å, respectively. Interesting deposited atomic models (PDBs PDB 6vsb, PDB 6vxx and PDB 6vyb) incompletely cover the reconstructed cryo-EM maps, showing the existence of disordered or over-sharpened regions after B-factor correction that could not be modelled. Supplementary Fig. 5 displays corresponding maps and fitted atomic models showing a large amount of protein that is not currently modelled.

In Fig. 5A, we show EMD-21375 map and the obtained LocSpiral reconstruction. In this figure, we use a relatively low threshold to visualise the outer parts of the protein. This figure shows that our obtained reconstruction presents less fragmented and broken densities and better map connectivity than the one deposited in EMD, suggesting that our approach improves the analysis and visualisation of the outer regions and potentially aides in the modelling of additional map motifs. In Supplementary Fig. 6A, we show similar results for EMD-21374, EMD-21452 and EMD-21457 maps. Interesting, the LocSpiral EMD-21374 map shows some additional fragmented densities at the top of the spike, however, we believe that these additional densities are in fact artefacts that come as a result of artificially imposing C3 symmetry on particles that are asymmetric. In Fig. 5B, we show the local B-factor map to be used for sharpening (slope of the local Guinier plot multiplied by 4) obtained by LocBFactor for EMD-21375 and in Supplementary Fig. 6B, we compare obtained local B-factor maps from EMD-21375, EMD-21374, EMD-21452 and EMD-21454 maps using a similar colourmap. Supplementary Fig. 6B shows that EMD-21452 and EMD-21454 present lower B-factors than EMD-21374 and EMD-21374, and then a better localizability of secondary structure and residues.

Then, we used the LocSpiral EMD-21375 reconstruction to improve the deposited atomic model (PDB 6vsb). As result, we could model additional loops and motifs: K444.C-F490.C; E96.C-S98.C; NAG1322.C; P812.C-K814.C, and some additional amino acids, which are now visible in the improved map: P621.C-G639.C; S673.C-V687.C; A829.B-A825.B. We were also able to visualise map densities corresponding to numerous additional N-linked glycans that could not be resolved in the original reconstruction. Examples of some regions that could be further modelled are shown in Fig. 5C, D. In Fig. 5C, we show the obtained LocSpiral map with the improved atomic model in green at the left and marked with an asterisk. At the right, it is rendered the deposited EMD map with the PDB 6vsb in magenta. Figure 5D shows in white the PDB 6vsb with the traced parts of the glycan proteins marked with purple spheres and in red the additional parts that could be traced using LocSpiral map. In addition, in this figure, we provide also zoomed views of two glycan proteins that could be further modelled with our improved map. Corresponding EMRINGER and MolProbity scores, calculated between LocSpiral map and the improved atomic model, and between EMD-21375 and the deposited model (PDB 6vsb), are shown in Supplementary Table S1. In both cases, the atomic structures were refined against corresponding maps by Phenix real_space_refine approach[25] using 5 refining iterations.

## Discussion

In this paper, we have introduced methods to improve the analysis and interpretability of cryo-EM maps. These methods include map enhancement approaches (LocSpiral and LocB-Sharpen), and approaches to calculate local B-factors (LocBFactor) and density occupancy maps (LocOccupancy). We have shown in our experiments that LocSpiral approach improves map connectivity showing fewer fragmented and broken densities and better coverage of the atomic model. In fact, our LocSpiral approach has been applied on several published publications[29–33],

enabling molecular modelling on maps with flexibility and light anisotropic resolution.

We envision that our proposed methods to estimate local B-factors and occupancy maps could be used to improve de novo model building. First, these maps can be employed to guide the manual tracing. These maps can be informative to estimate the range of structures that could be compatible with the given electron microscopy density. Second, for very high-resolution cryo-EM maps, these values can be used as an approximation of the atomic B-factors and occupancies to be further refined as part of the automatic model refinement process by automatic model building packages as Phenix[14] or Refmac[15]. B-factor maps provide complementary information to local resolution maps, though, these results are usually correlated. The latter usually determines the resolution at a given point by comparing the map to noise or background amplitudes[34], while the former determines the rate of signal amplitude fall off within a resolution range. Then, we can find map regions with similar local resolution (map amplitude similar to noise/background amplitude at this resolution and coordinates), while different B-factor as the signal damping could be different within the used resolution range (highly or slowly sloped).

We have seen that we must be careful when processing maps affected by high flexibility and heterogeneity or when analysing maps with moderate global resolution (close to 10–15 Å) as the obtained B-factors could be overestimated if the selected resolution range is above the local resolution at these regions. Note that obtained B-factors at this low-resolution regions describe mainly noise B-factors that show how the noise signal fall off inside the used resolution range and they should be filtered out from our B-factor map. However, these problematic cases can be easily detected as the amplitude values in corresponding Guinier plot will be below the noise level (obtained from the 90–95% quantile of the empirical noise/background distribution). Thus, these regions can be automatically filtered out and not taken into consideration. In our analysis of B-factors for low and high-resolution maps shown in the Supplementary Material, we show that existing methods to determine the map global B-factor, as Relion postprocessing, do not filter problematic low-resolution regions so the estimated B-factor may be overestimated.

In principle, it might be possible to differentiate between compositional and moderate conformational flexibility from the obtained occupancy maps for samples accurately 3D classified. In the former case, the occupancy map is expected to show close to zero values at missing regions, as the density values of these parts should be low and close to the noise level. Oppositely, in the latter case, the occupancy is likely to show higher values as the density values of moving parts, while slightly blurred because of the movement, should be similar to the ones at other static regions of the macromolecule. However, we should be extremely careful about these analyses as 3D classification approaches are not perfect, thus, macromolecules showing different compositions could provide 3D maps with significant density values in regions that should be empty. Additionally, samples showing large conformational changes could present low-density values at moving regions when compared to density values at static parts, providing close-to zero occupancy values.

The methods proposed here are semi-automated and essentially only require the unfiltered map to enhance or analyse, a resolution range and, in some cases, a binary solvent mask as inputs. They do not require additional information as atomic models or local resolution maps. The common link between all these approaches is the use of the spiral phase transform, which is used to factorise cryo-EM maps into amplitude and phase terms in real space for different resolutions. The spiral phase transform has been extensively used in optics for phase extraction in

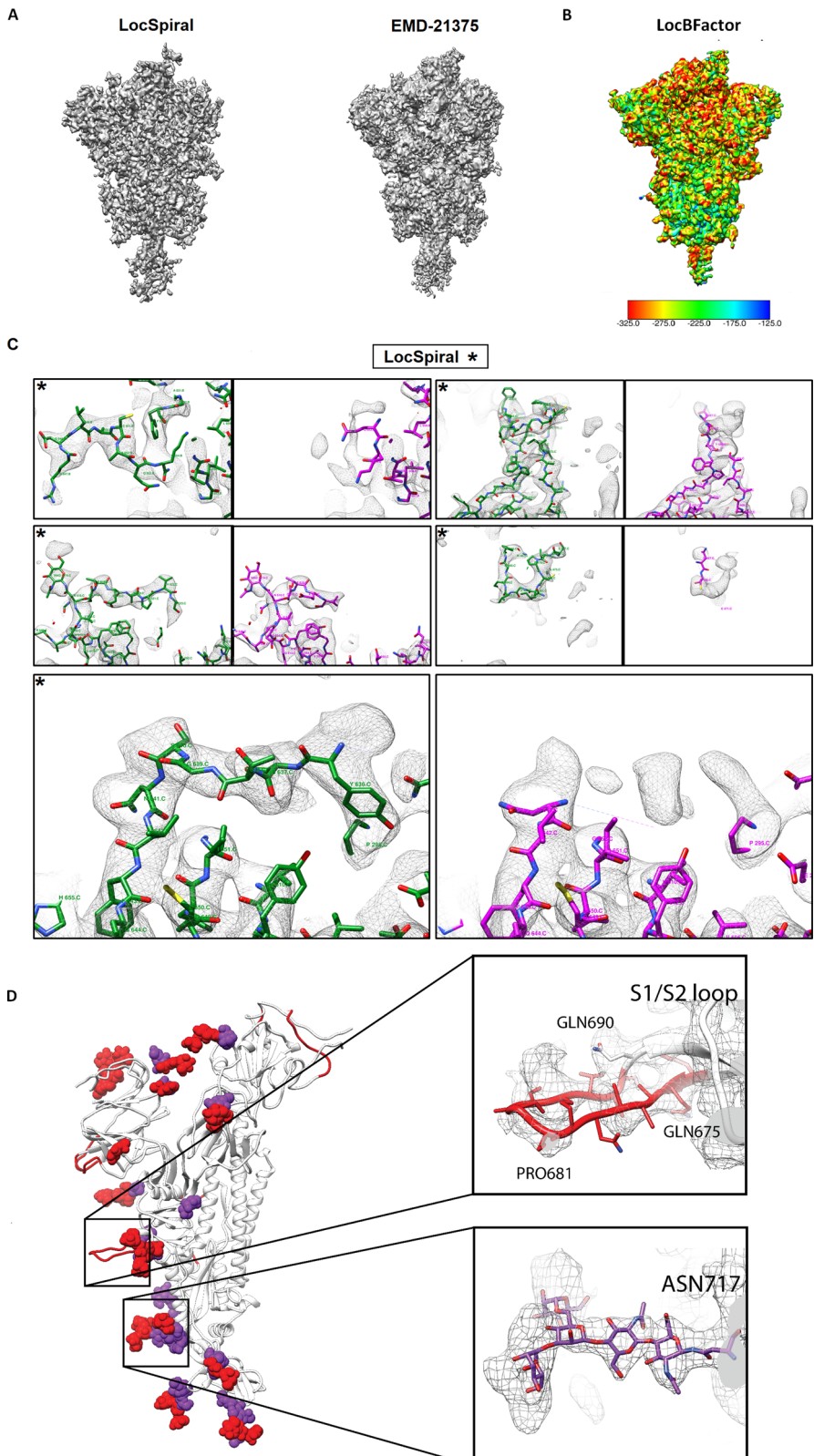

**Fig. 5 Results obtained by LocSpiral, LocBFactor and improved atomic model for EMD-21375 SARS-CoV-2 sample. A** Map obtained by LocSpiral approach (left) compared with the map as deposited in EMDB for EMD-21375. **B** *B*-factor maps to be used for sharpening (slope of the local Guinier plot multiplied by 4) obtained by LocBFactor approach for EMD-21375. The *B*-factor ranges between [−325.0, −125.0] Å$^2$. **C** Visual examples of map regions corresponding to EMD-21375 that could be further modelled after processing the corresponding unfiltered and unsharpened map with LocSpiral approach. On the left and marked with asterisks, we show the LocSpiral maps with the improved atomic models in green, and on the right the deposited EMD-21375 map with the PDB 6vsb in magenta. **D** In white, PDB 6vsb with traced parts of the glycan proteins marked with purple spheres. In red, additional parts that could be traced using LocSpiral map. Inside the black squares, zoomed views of two glycan proteins that could be further modelled.

interferometry[35–39] or by Shack-Hartmann sensors[40,41]. This transformation is not new in cryo-EM as it has been proposed previously to facilitate particle screening[42], CTF estimation[43] and local and directional resolution determination[34,44]. In refs. [34,44], the authors used the Riesz transform to obtain amplitude maps, which is similar to the spiral phase transform.

Cryo-EM reconstructions of different types of macromolecules have been used to test the performance of these algorithms. Specifically, we have used a membrane protein (TRP channel), immature ribosomes affected by high compositional heterogeneity, the spliceosome that shows high conformational heterogeneity, recent SARS-CoV-2 reconstructions exhibiting dynamic regions and high-resolution apoferritin reconstructions. In all cases, our proposed approaches show excellent results, improving the analysis and the interpretability of the processed maps. The proposed methods are also highly efficient. For example, the processing of EMD-21457 (map size 400 px$^3$) using our local enhancement approach took only 12 min on a standard laptop using 4 cores.

## Methods

The proposed methods are based on a 3D generalisation of the 2D spiral phase transform. In the following, we present the 3D spiral phase transform and its application to map enhancement, local *B*-factor determination, and estimation of local map occupancies.

**3D spiral phase transform**. The spiral phase transform is a Fourier operator that can factorise a 3D map into its amplitude and phase terms in real space at different resolutions. We assume without loss of generality that a given 3D map can be modelled as a 3D phase modulated signal given by

$$V(\mathbf{r}) = \sum_{\omega} V_{\omega}(\mathbf{r}) = \sum_{\omega} \left( b_{\omega}(\mathbf{r}) + m_{\omega}(\mathbf{r})\cos(\varphi_{\omega}(\mathbf{r})) \right) \tag{3}$$

where $V(\mathbf{r})$ is the cryo-EM map, $V_{\omega}(\mathbf{r})$ is a band-passed map filtered at frequency $\omega$, $b_{\omega}(\mathbf{r})$ the 3D background or DC term, $m_{\omega}(\mathbf{r})$ the 3D amplitude map, $\varphi_{\omega}$ the 3D modulating phase and $\mathbf{r} = (x, y, z)$. Assuming that we are interested in spatial frequencies higher than 1/50–1/30 1/Å and that the background is usually a low-frequency signal, we can approximate the map by a high-passed filtered map $V_{HP}$ for resolutions higher than 50–30 Å by

$$V_{HP}(\mathbf{r}) = \sum_{\omega} m_{\omega}(\mathbf{r})\cos(\varphi_{\omega}(\mathbf{r})) \tag{4}$$

For convenience, Eq. (4) can be expanded into its corresponding analytic signal as

$$\tilde{V}_{HP}(\mathbf{r}) = \sum_{\omega} m_{\omega}(\mathbf{r})e^{j\varphi_{\omega}(\mathbf{r})} \tag{5}$$

This analytic signal relates to our high-passed filtered map by

$$V_{HP}(\mathbf{r}) = \sum_{\omega} \text{Re}\{\tilde{V}_{HP}(\mathbf{r})\} = \sum_{\omega} \text{Re}\{m_{\omega}(\mathbf{r})e^{j\varphi_{\omega}(\mathbf{r})}\} =$$
$$V_{HP}(\mathbf{r}) = \sum_{\omega} \text{Re}\{(m_{\omega}(\mathbf{r})\cos(\varphi_{\omega}(\mathbf{r})) + jm_{\omega}(\mathbf{r})\sin(\varphi_{\omega}(\mathbf{r})))\} \tag{6}$$

with $\text{Re}\{\cdot\}$ an operator that takes the real part and $j$ is the imaginary unit ($j^2 = -1$). Note from the analytic signal defined in Eq. (5) that $m_{\omega}(\mathbf{r})$ and $\varphi_{\omega}(\mathbf{r})$ clearly represent amplitude and phase terms. The quadrature transformation of Eq. (4) is given by

$$Q\{V_{HP}(\mathbf{r})\} = -\sum_{\omega} m_{\omega}(\mathbf{r})\sin(\varphi_{\omega}(\mathbf{r})) \tag{7}$$

Then, Eq. (5) may be rewritten as

$$\tilde{V}_{HP}(\mathbf{r}) = \sum_{\omega} (V_{HP}(\mathbf{r}) - jQ\{V_{HP}(\mathbf{r})\}) \tag{8}$$

Assuming that $m_{\omega}$ is a low varying map compared to $\varphi_{\omega}$, the gradient of $V_{HP}$ is approximated by

$$\nabla V_{HP}(\mathbf{r}) \cong -\sum_{\omega} m_{\omega}(\mathbf{r})\sin(\varphi_{\omega}(\mathbf{r}))\nabla\varphi_{\omega}(\mathbf{r}) \tag{9}$$

Rearranging terms, we obtain

$$Q\{V_{HP,\omega}(\mathbf{r})\} = \frac{\nabla\varphi_{\omega}(\mathbf{r})}{|\nabla\varphi_{\omega}(\mathbf{r})|} \cdot \frac{\nabla V_{HP,\omega}(\mathbf{r})}{|\nabla\varphi_{\omega}(\mathbf{r})|} = -\mathbf{n}_{\varphi}(\mathbf{r}) \cdot \frac{\nabla V_{HP,\omega}(\mathbf{r})}{|\nabla\varphi_{\omega}(\mathbf{r})|} \tag{10}$$

Equation (10) shows that the quadrature term is composed of two terms. The first is an orientation map $\mathbf{n}_{\varphi}$ and the second corresponds to a non-linear operator that can be interpreted as a 3D generalisation of the 1D Hilbert transform, which can be efficiently calculated using the Fourier transform. As shown in[45], the

operator $\nabla V_{HP,\omega}(\mathbf{r})/|\nabla\varphi_{\omega}(\mathbf{r})|$ corresponds to the 3D Hilbert transform applied to our band-passed maps $V_{HP,\omega}(\mathbf{r})$, then

$$\mathbf{H}\{V_{HP,\omega}(\mathbf{r})\} = \text{FT}^{-1}\left\{\frac{-i\mathbf{q}}{\mathbf{q}}\text{FT}\{V_{HP,\omega}(\mathbf{r})\}\right\} \cong \frac{\nabla V_{HP,\omega}(\mathbf{r})}{|\nabla\varphi_{\omega}(\mathbf{r})|} \tag{11}$$

Thus, Eq. (10) can be rewritten as

$$Q\{V_{HP,\omega}(\mathbf{r})\} = \frac{\nabla\varphi_{\omega}(\mathbf{r})}{|\nabla\varphi_{\omega}(\mathbf{r})|} \cdot \frac{\nabla V_{HP,\omega}(\mathbf{r})}{|\nabla\varphi_{\omega}(\mathbf{r})|} \cong -\mathbf{n}_{\varphi}(\mathbf{r}) \cdot \text{FT}^{-1}\left\{\frac{-i\mathbf{q}}{\mathbf{q}}\text{FT}\{V_{HP,\omega}(\mathbf{r})\}\right\} \tag{12}$$

Note that $\mathbf{n}_{\varphi}$ is a unit vector pointing in the same direction that $\nabla V_{HP,\omega}(\mathbf{r})$ (remember that $m_{\omega}$ is a low varying map compared to $\varphi_{\omega}$), but maybe with different orientation because a possible change of sign introduced by the cosine term in Eq. (4). We can rewrite Eq. (12) as

$$Q\{V_{HP,\omega}(\mathbf{r})\} \cong -\mathbf{n}_{\varphi}(\mathbf{r})\left|\text{FT}^{-1}\left\{\frac{-i\mathbf{q}}{|\mathbf{q}|}\text{FT}\{V_{HP,\omega}(\mathbf{r})\}\right\}\right|\mathbf{n}_{V_{HP,\omega}}(\mathbf{r})$$
$$= -s(\mathbf{r})\left|\text{FT}^{-1}\left\{\frac{-i\mathbf{q}}{|\mathbf{q}|}\text{FT}\{V_{HP,\omega}(\mathbf{r})\}\right\}\right| \tag{13}$$

where $s(\mathbf{r})$ is a function with range $+1$ or $-1$ considering that $\mathbf{n}_{\varphi}(\mathbf{r})$ and $\mathbf{n}_{V_{HP,\omega}}$ can be parallel or antiparallel only. From Eq. (13), we can obtain an estimation of $\varphi_{\omega}(\mathbf{r})$ affected by an indetermination in its sign by

$$\varphi_{\omega}(\mathbf{r}) \cong \arctan\left[\frac{Q\{V_{HP,\omega}(\mathbf{r})\}}{V_{HP,\omega}(\mathbf{r})}\right] = -s(\mathbf{r})\arctan\left[\frac{|\text{FT}^{-1}\{\frac{-i\mathbf{q}}{|\mathbf{q}|}\text{FT}\{V_{HP,\omega}(\mathbf{r})\}\}|}{V_{HP,\omega}(\mathbf{r})}\right] \tag{14}$$

However, we can use Eq. (14) to obtain the modulation and cosine terms in Eq. (4) separately without sign ambiguity as

$$\cos(\varphi_{\omega}(\mathbf{r})) = \cos\left(\arctan\left[\frac{\text{Im}\{\tilde{V}_{HP}(\mathbf{r})\}}{\text{Re}\{\tilde{V}_{HP}(\mathbf{r})\}}\right]\right) \cong \cos\left(\arctan\left[\frac{|\text{FT}^{-1}\{\frac{-i\mathbf{q}}{|\mathbf{q}|}\text{FT}\{V_{HP,\omega}(\mathbf{r})\}\}|}{V_{HP,\omega}(\mathbf{r})}\right]\right)$$
$$m_{\omega}(\mathbf{r}) = (\tilde{V}_{HP}(\mathbf{r}) \cdot \tilde{V}_{HP}(\mathbf{r})^+)^{1/2} = \left((V_{HP,\omega}(\mathbf{r}))^2 + (Q\{V_{HP,\omega}(\mathbf{r})\})^2\right)^{1/2} \tag{15}$$

Using these expressions, we can obtain for each frequency $\omega$ the terms $\cos(\varphi_{\omega}(\mathbf{r}))$ and $m_{\omega}(\mathbf{r})$.

**Local enhanced map (LocSpiral)**. We are proposing here a robust local map enhancement method that only requires as input a binary mask of the macromolecule and a resolution range. The approach works for both high and moderate resolution maps. In the following, we provide details of the proposed method.

As explained before, each band-pass filtered map can be factorised into an amplitude and phase term by the spiral phase transform. Then, given a user-defined solvent mask, the method obtains the empirical noise amplitude probability distribution ($m_{\omega}^N$) at frequency ω, selecting the density values of voxels not included in the solvent mask. From this distribution, the approach determines the noise amplitude value corresponding to the 90–95% quantile, given by $m_{\omega}^N(q = 95\%)$. This value is used to locally normalise map amplitudes in real space along with different frequencies and remove local signals that are below this amplitude threshold as they are likely noise at this given frequency and position. After this non-linear amplitude transformation, the enhanced map at a given frequency ω is given by

$$\breve{V}_{\omega}(\mathbf{r}) = (m_{\omega}(\mathbf{r}) > m_{\omega}^N(q = 95\%))\cos(\varphi_{\omega}(\mathbf{r})) \tag{16}$$

and the map

$$\breve{V}(\mathbf{r}) = \sum_{\omega} \breve{V}_{\omega}(\mathbf{r}) = \sum_{\omega} (m_{\omega}(\mathbf{r}) > m_{\omega}^N(q = 95\%))\cos(\varphi_{\omega}(\mathbf{r})) \tag{17}$$

The method allows as an option the use of an SNR weighting parameter to weight the contribution of the different amplitudes in the final map. In this case, Eq. (17) is rewritten as

$$\breve{V}(\mathbf{r}) = \sum_{\omega} C_{\text{ref},\omega}(\mathbf{r})(m_{\omega}(\mathbf{r}) > m_{\omega}^N(q = 95\%))\cos(\varphi_{\omega}(\mathbf{r})) \tag{18}$$

with $C_{\text{ref},\omega}(\mathbf{r})$ the SNR weighting parameter given by

$$C_{\text{ref},\omega}(\mathbf{r}) = \frac{m_{\omega}(\mathbf{r})}{m_{\omega}(\mathbf{r}) + m_{\omega}^N(q = 95\%)} \tag{19}$$

**Local *B*-factor determination (LocBFactor)**. The factorisation of a 3D map into its amplitude and phase terms in real space for different frequencies allows the efficient determination of local *B*-factor maps. To this end, LocBFactor method first obtains the local map amplitudes $m_{\omega}(\mathbf{r})$ for resolutions between 15–10 Å to the estimated global map resolution. These amplitude maps are then used to obtain SNR-weighted log-amplitudes of structure factors locally as

$$\log(F_{\omega}(\mathbf{r})) = \log\left(C_{\text{ref},\omega}(\mathbf{r})m_{\omega}(\mathbf{r})\right) \tag{20}$$

with $C_{\text{ref},\omega}(\mathbf{r})$ a SNR weighting parameter defined in (19). This expression can be used to fit $\log(F_\omega(\mathbf{r}))$ versus $\omega^2$ within the resolution rage defined between 15 and 10 Å to the estimated global map resolution. Thus, finally we have

$$\log(F_\omega(\mathbf{r})) \cong B(\mathbf{r})(\omega^2 - \omega_0^2) + A(\mathbf{r}) \tag{21}$$

with $B(\mathbf{r})$ the local $B$-factor map or B map, and $A(\mathbf{r})$ the log-amplitude map at $\omega_0$ (A map). In Eq. (21) the approach typically does not take into consideration in the linear fit amplitude values $(m_\omega(\mathbf{r}))$ that are below the noise level $(m_\omega^N(q= 95\%))$. Additionally, local Guinier plots without at least two points above the noise level are filtered out from the B map. Note that $\omega_0$ corresponds to the lowest frequency within the used resolution range (typically $1/15$–$1/10$ Å$^{-1}$).

**Local $B$-factor sharpened map (LocBSharpen)**. The spiral phase transform can be used to obtain local $B$-factor sharpened maps. Note that Expression (4) can be modified for frequencies higher than $\omega_0$ as

$$\breve{V}(\mathbf{r}) = \sum_\omega \breve{V}_{\text{HP},\omega}(\mathbf{r}) = \begin{cases} \sum_\omega \left( C_{\text{ref},\omega}(\mathbf{r}) m_\omega(\mathbf{r}) \cos(\varphi_\omega(\mathbf{r})) \right), & \omega < \omega_0 \\ \sum_\omega \left( C_{\text{ref},\omega}(\mathbf{r}) A(\mathbf{r}) \cos(\varphi_\omega(\mathbf{r})) \right), & \omega \geq \omega_0 \end{cases} \tag{22}$$

With $A(\mathbf{r})$ the log-amplitude map at $\omega_0$ (A map).

**Local occupancy map (LocOccupancy)**. Low occupancy map regions correspond to parts of the macromolecule where map amplitudes of the reconstruction are significantly smaller when compared to other regions of the macromolecule. Keeping this in mind, we define the occupancy map as

$$O(\mathbf{r}) = \frac{\sum_\omega \left( m_\omega(\mathbf{r}) > m_\omega^M(q = 25\%) \right)}{\sum_\omega \left( m_\omega(\mathbf{r}) \geq m_\omega^M(q = 0\%) \right)} \tag{23}$$

where $m_\omega^M(q= 25\%)$ and $m_\omega^M(q = 0\%)$ are obtained from the empirical macro-molecule amplitude probability distribution $(m_\omega^M)$ at frequency ω. This amplitude probability distribution is calculated from map density values corresponding to voxels that are included in the solvent mask. From this distribution, the approach determines the macromolecule amplitude values corresponding to the 25 and 0% quantiles, given by $m_\omega^M(q = 25\%)$ and $m_\omega^M(q= 0\%)$ that are used as thresholds. To calculate local occupancy maps, a typical resolution range between 30 and 10–8 Å is used to obtain density occupancies of complete secondary structure motifs, while ranges between 5 and 3–1.5 Å are used for high-resolution cryo-EM maps to obtain occupancies of residues.

**Maturity level index**. In the analysis of the immature 50S ribosomes, we have proposed a maturity level index. This index can be extended to the analysis of any maturing macromolecule and is useful to place immature macromolecules into a maturing timeline. The calculation of this index requires reconstructions of immature and mature macromolecules. The mature reconstruction is used to obtain a binary solvent mask, while the immature reconstructions are used to calculate occupancy maps. These occupancy maps allow us to determine highly occupied regions (occupancy >0.75) and calculate occupancy masks. Then, the index is obtained comparing the number of voxels activated in the solvent mask of the mature reconstruction with the ones in the occupancy masks. As can be seen from Fig. 3, the larger are the regions that are not folded in the immature maps, the smaller is the maturity level.

**Cryo-EM image processing of the spliceosome data**. The dataset is composed of 327,490 particle images of a spliceosomal B-complex from yeast (EMPIAR-10180)[4]. The particles were polished with Relion, downsampled to 1.699 Å/px and windowed to a size of 320 × 320 pixels. A set of 30 initial volumes were obtained by RANSAC (15 maps) and Eman2 (15 maps) and processed by volume selector approach[22] producing two different initial volumes. Then, Relion 3D classification was used to compute two classes providing both volumes as reference initial maps (class 1 and class 2 composed by 201,407 and 126,083 particles respectively). The resulting classes were refined by Relion autorefine using the maps obtained in the previous 3D classification. Finally, Relion postprocessing provided maps at 4.28 and 4.58 Å for class 1 and class 2, respectively. Lastly, a local resolution was calculated using Relion for both classes.

## Data availability
Previously published datasets used for testing are available from the Electron Microscopy Data Bank (https://www.ebi.ac.uk/pdbe/emdb/) under accession codes EMD-10418, EMD-8440, EMD-8441, EMD-8445, EMD-8450, EMD-8434, EMD-21375, EMD-21374, EMD-21452 and EMD-21457. Data that support the findings of this study have been deposited in http://t.ly/XKQa.

## Code availability
The source code for the presented methods is freely available under the terms of an open-source software license and can be downloaded from https://github.com/1aviervargas/LocSpiral-LocBSharpen-LocBFactor-LocOccupancy[46].

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

## Acknowledgements

This work was supported by grants from NSERC Discovery Grant (RGPIN-2018-04813), the Spanish Ministry of Science and Innovation through the call 2019 Proyectos de I + D + i - RTI Tipo A (PID2019-108850RA-I00). J.V. acknowledges economical support from the Ramón y Cajal 2018 programme (RYC2018-024087-I). We want to thank helpful discussions with Jose Jesus Fernandez.

## Author contributions

J.V. had the idea, J.V., S.K. and J.G.-B. and devised the theory, developed and implemented the algorithm, performed the experiments and wrote the manuscript. R.S.-G. and S.A. helped to analyse and interpret data. A.A.Z.K., D.W., J.S.M. and K.H.B. analysed data, wrote part of the manuscript and provided comments and feedback. All authors reviewed the manuscript, supervised the experiments and discussed the results.

## Competing interests

The authors declare no competing interests.
