## [Peer Review File · Nature Communications]

REVIEWER COMMENTS

Reviewer #1 (Remarks to the Author):

The work entitled "Local computational methods to improve the interpretability and analysis of cryo-EM maps" provides a new approach to estimate the local resolution and make corresponding sharpening for a cryoEM density map. This is very important to visualize the structural information from a given map without any extra requirements, such as a pre-fitted model which may lead to model bias. The testing results is very exciting, and demonstrates clear advantages than other exist methods, which is an essential progress in the field.

- 1) The occupancy is usually related to particles with missing sub-regions, temporarily called compositional flexibility. The present work used an intensity criterion to evaluate the occupancy. The intensity of a density might also be influenced by flexibility, temporarily called conformational flexibility. How to distinguish the compositional flexibility from the conformational one? This should be explained.
- 2) In Fig 1A, the middle panel of overlap maps, the presentation should be improved to enhance the differences.
- 3) In Fig 3 C, the result from LocSpiral contains more extra densities than the red map from Relion. The author shows such densities, but not explains what they are. Densities from missing residues or noise?

Reviewer #2 (Remarks to the Author):

Summary:

CryoEM reconstructions require amplitude scaling in order to effectively represent densities across resolution shells. To date, most cryoEM reconstructions only consider a single B-factor that is used to scale amplitudes for all regions of a reconstruction, which typically results in some regions having correct scaling and other areas having inappropriate scaling (i.e. floating, unconnected densities). To deal with this, users will typically sharpen maps at different resolutions and B-factors to enable model building, which requires significant efforts and expertise in determining what the 'correct' B-factor should be for a given part of a map.

Overall, this paper represents a novel approach to obtaining locally sharpened reconstructions in a manner that is user-independent (i.e. does not require subjective optimizations) and therefore is of general interest to the cryoEM & structural biology community.

To be addressed in the revision:

-Reported B-factor value for B-maps from LocalBfactor. I am confused about why the values are reversed compared to typical B-factors reported by cryoEM software. For example, in Figure 2D, the regions with low occupancy and lower resolution have lower B-factor values compared to the core of the spliceosome. This is counter to how this is reported currently and is a confusing aspect to this software. To address this point of confusion, I request the authors have an explicit discussion of how to relate these values to current B-factor values or to 'invert' the values to make the reported values from LocalBfactor consistent with existing conventions.

-Modeling statistics: The authors include modeling statistic comparisons for the TRP channel but not for reconstructions. I think these values should be generated for atomic-resolution maps such as apoferritin and the spike protein so that readers can easily see whether this new software improved modeling statistics. Right now, the reader is left unknown as to whether improvements that the authors saw for the TRP channel extend to other maps (it appears that it will, but having modeling statistics will confirm this). This is particularly important for the new model building performed on the spike protein; a 'Table 1' description needs to be provided for modeling

information and statistics.

-The authors performed single particle analysis for the spliceosome. As such, for rigor and transparency the authors should provide a description in the methods sections regarding data processing.

--Michael Cianfrocco, PhD

**Answer to referee comments on:
"Local Computation methods to improve the
interpretability and analysis of cryo-EM maps"**

Manuscript ID NCOMMS-20-26198A-Z

To Whom It May Concern,

We thank the reviewers for their constructive comments and the editor for overseeing the review. Please find enclosed our response to the referee comments.

%%%

Reviewer #1:

The work entitled "Local computational methods to improve the interpretability and analysis of cryo-EM maps" provides a new approach to estimate the local resolution and make corresponding sharpening for a cryoEM density map. This is very important to visualize the structural information from a given map without any extra requirements, such as a pre-fitted model which may lead to model bias. The testing results is very exciting, and demonstrates clear advantages than other exist methods, which is an essential progress in the field.

1) *The occupancy is usually related to particles with missing sub-regions, temporarily called compositional flexibility. The present work used an intensity criterion to evaluate the occupancy. The intensity of a density might also be influenced by flexibility, temporarily called conformational flexibility. How to distinguish the compositional flexibility from the conformational one? This should be explained.*

We agree with the reviewer. The proposed method estimates the local occupancy analyzing the intensity values of the electron density map. In principle, it could be possible to differentiate between compositional and moderate conformational flexibility from occupancy maps. In the first case, the occupancy map is expected to show close to zero values in the missing regions, as the density values of these parts should be low and close to the noise level after classifying the particle set. Oppositely, the occupancy is likely to show higher values for maps affected by small conformational flexibility. In this case, the density values of the moving parts, while slightly blurred because of the movement, should be similar to the ones at other static regions of the macromolecule inside the used resolution range.

In practice, however, the situation is more complicated. In first place, 3D classification approaches are not perfect, thus, macromolecules showing different compositions could provide 3D maps with significant density values in regions that should be empty. In addition, cryo-EM samples showing large conformational changes could present very

low-density values at moving regions when compared to density values at static parts of the macromolecule. This happens because the intensity is spread out along very large trajectories. In these two cases, the occupancy values that we will obtain will be opposed to the analysis made previously.

The occupancy examples shown in the manuscript are helpful to understand these different cases. On the one hand, the results obtained for the immature prokaryote ribosomes and the apoferritin aligns well with the first analysis made above. The occupancy maps shown in Figure 4 indicate that immature ribosomes present occupancy values close to zero in not folded regions. Moreover, for the apoferritin, the occupancy map shown in Figure 3 reveals occupancies as low as approximately 0.5 at the outer part, indicating the presence of a small conformational flexibility in the outer residues. On the other hand, results obtained for the PC2 TRP channel and the spliceosome are more problematic. In Figure S2 (C), we show the obtained local occupancy map for the channel. Interesting, this map shows low occupancies at regions occupied by detergent and cholesterol densities (please see Figure 2 in (Wang, Corey et al. 2020)), suggesting the presence of compositional variability in these regions. However, the obtained occupancy values are close to 0.5, indicating that there is a mixture of particles not efficiently classified that show compositional heterogeneity at these regions. Additionally, the spliceosome results show occupancy values close to zero for the flexible and moving parts (Figure 2C). These low occupancy values are consequence of the large movements experienced by the helicase and SF3b domains as discussed above.

To clarify this point, we have included the following paragraph in the Discussion section of the manuscript:

“In principle, it might be possible to differentiate between compositional and moderate conformational flexibility from the obtained occupancy maps for samples accurately 3D classified. In the former case, the occupancy map is expected to show close to zero values at missing regions, as the density values of these parts should be low and close to the noise level. Oppositely, in the latter case, the occupancy is likely to show higher values as the density values of moving parts, while slightly blurred because of the movement, should be similar to the ones at other static regions of the macromolecule. However, we should be extremely careful about these analyses as 3D classification approaches are not perfect, thus, macromolecules showing different compositions could provide 3D maps with significant density values in regions that should be empty. Additionally, samples showing large conformational changes could present low-density values at moving regions when compared to density values at static parts, providing close-to zero occupancy values.”

%%%

2) In Fig 1A, the middle panel of overlap maps, the presentation should be improved to enhance the differences.

Following the referee indication, we have included a new figure (Figure S1) where we improve the comparison between LocSpiral and Relion maps for the TRP channel. Note that the aim of the middle panel of Fig 1A is to show that the density maps for both cases are similar in the inner core of the protein at the working density thresholds. In the new Figure S1 (A), we show the complete and overlapping LocSpiral and Relion maps with the corresponding atomic structure. Additionally, Figure S1 (B) shows reconstructions of corresponding regions at the core and bottom outer region of the TRP channel obtained from LocSpiral (left) and Relion (right) approaches. As can be seen from these figures, the map densities are similar in the inner core of the protein (actually, it seems that the map density threshold used to render LocSpiral is higher than the one used to show Relion map). However, the map densities are quite different at the outer regions, where the Relion map shows thin and broken densities.

For the sake of clarity, we reproduce this new figure here:

Figure S1 Comparison between LocSpiral and Relion postprocess maps for the TRP channel. A) Complete and overlapping LocSpiral and Relion maps shown with the

corresponding atomic structure (PDB 6t9n), B) Reconstructions of corresponding regions at the core and bottom outer region of the TRP channel obtained from LocSpiral (left) and Relion (right) approaches.

%%

3) In Fig3 C, the result from LocSpiral contains more extra densities than the red map from Relion. The author shows such densities, but not explains what they are. Densities from missing residues or noise?

We agree with the referee and we have added a new figure to show that these extra densities correspond to missing residues. To this aim, we have fitted the apoferritin atomic structure (PDB ID 6v21) to the reconstructed maps. The results are given in new Figure S4. For the sake of clarity, we reproduce here this new figure.

Figure S4 Complete and superimposed sharpened maps obtained by LocSpiral (gray colour) and Relion (red colour) for EMD-9865 with the corresponding atomic structure (PDB 6v21). In the black rectangles are shown zoomed views of the regions labelled with the same index.

Additionally, we have included the following sentence in the Apoferritin subsection:

“Figure S4 shows that the extra densities that appear in the LocSpiral map correspond to missing residues in EMD-9865.”

%%

Reviewer #2:

Summary:

CryoEM reconstructions require amplitude scaling in order to effectively represent densities across resolution shells. To date, most cryoEM reconstructions only consider a single B-factor that is used to scale amplitudes for all regions of a reconstruction, which typically results in some regions having correct scaling and other areas having inappropriate scaling (i.e. floating, unconnected densities). To deal with this, users will typically sharpen maps at different resolutions and B-factors to enable model building, which requires significant efforts and expertise in determining what the 'correct' B-factor should be for a given part of a map.

Overall, this paper represents a novel approach to obtaining locally sharpened reconstructions in a manner that is user-independent (i.e. does not require subjective optimizations) and therefore is of general interest to the cryoEM & structural biology community.

To be addressed in the revision:

1) *Reported B-factor value for B-maps from LocalBfactor. I am confused about why the values are reversed compared to typical B-factors reported by cryoEM software. For example, in Figure 2D, the regions with low occupancy and lower resolution have lower B-factor values compared to the core of the spliceosome. This is counter to how this is reported currently and is a confusing aspect to this software. To address this point of confusion, I request the authors have an explicit discussion of how to relate these values to current B-factor values or to 'invert' the values to make the reported values from LocalBfactor consistent with existing conventions.*

We thank the referee for pointing out this point of confusion. We want to mention that in the previous manuscript, this behavior where regions with low resolutions show lower B-factors compared with other regions with higher resolutions only happened with highly heterogeneous maps: Spliceosome and immature ribosome. For the rest of maps (TRP channel, Apoferritin and SARS-CoV-2), the obtained B-factor maps correlate well with corresponding local resolution maps. Please see Figure S2 in [1], Figure 2 in [2] and Figure S2 in [3], for example. However, we totally agree with the reviewer and we believe that this point of confusion can introduce misleading interpretations of local B-factors when processing maps affected by high flexibility and heterogeneity, or when analyzing maps with moderate global resolution (close to 10-15 Å). Thus, we have decided to introduce a new option in LocBFactor method to avoid this potential problem.

The cause of this point of confusion is that the resolution range used in the calculation of the local B-factors is above the local resolutions in the flexible and low-resolution parts of the Spliceosome and the immature ribosome. Note that Guinier plots at regions with amplitudes below the noise level (obtained from the 90-95% quantile of the empirical noise/background distribution) are dominated by the noise signal and describe the noise signal fall off inside the used resolution range. Consequently, the obtained B-factors at these low-resolution regions describe mainly noise B-factors that show how the noise signal fall off inside the used resolution range. The noise signal presents typically an

approximately flat spectrum, thus, artefactual close to zero B-factors, which are not in agreement with the concept of B-factor as a measure of position uncertainty or disorder.

The current version of LocBFactor can effectively filter out these noise B-factors from the B-factor map. To this end, local Guinier plots without at least two points above the noise level (obtained from the 90-95% quantile of the empirical noise/background distribution) are filtered out from the B map. Additionally, the approach does not take into consideration in the linear fit amplitude values that are below the noise level. Note that in the current manuscript version, the obtained B-factor values for the Spliceosome and immature ribosome do not look like reversed compared to typical B-factors reported by cryoEM software and correlated perfectly with corresponding A maps and local resolution maps (please see Figure 2E and F, and Figure S4 in the manuscript and Figure S4 Class 3 in [4]).

Importantly, existing methods to determine the map global B-factor, i.e. Relion postprocessing, do not filter problematic low resolution regions in the B-factor calculation. Thus, the estimated B-factor may be overestimated. In the updated manuscript, we have introduced a new section ‘B-factor analysis of low and high resolution maps’ in the Supplementary Material. In this new section, we process approximately homogeneous low- and high-resolution maps corresponding to EMD-20671 and EMD-21024. We show that Relion postprocessing provides an artefactual low B-factor value (-97.70 \AA^2) when processing a low-resolution map as EMD-20671, which is at 16.01 \AA resolution. Note that this B-factor value is similar to that of EMD-21024 (-50.81 \AA^2), which shows a very high global resolution of 1.77 \AA . When we process these maps by LocBFactor, we obtain that the average of signal B-factors inside respective solvent masks are -1172 \AA^2 (EMD-20671) and -78 \AA^2 (EMD-21024). Note that the values obtained by Relion and LocBFactor are similar for EMD-21024. Oppositely, the average signal B-factor obtained by LocBFactor for EMD-20671 is much lower and consistent with a map at 16.01 \AA resolution than the one reported by Relion. We believe that the reason of this discrepancy is because LocBFactor filters out noise B-factors (B-factors obtained from amplitudes below the noise level for the used resolution range) while Relion does not filter regions dominated by noise within the used resolution range.

As summary to clarify this point, we have made the following changes:

- 1) We have modified LocBFactor approach to filter out noise B-factors.
- 2) We have modified the Spliceosome section and Figure 2 to explain that noise B-factors should be filtered out from B-factor maps and show the new results.
- 3) We have modified the section ‘Immature prokaryote ribosomes’ in Supplementary material and Figure S4 with the new results obtained by LocBFactor for this heterogeneous sample.
- 4) We have introduced a new section in the Supplementary material where we have processed approximately homogeneous low- and high-resolution maps corresponding to

EMD-20671 and EMD-21024, and we have compared the results obtained by LocBFactor and Relion postprocessing methods. We have also included a new figure showing the corresponding results (Figure S7).

5) We have introduced a new paragraph in the Discussion section where we have summarized the ideas presented above.

[1] Wang et al., “Lipid Interactions of a Ciliary Membrane TRP Channel: Simulation and Structural Studies of Polycystin-2”, *Structure* 28(2) 169-184 e165 (2020)

[2] Wu et al., “Sub-2 Angstrom resolution structure determination using single-particle cryo-EM at 200 keV”, *Journal of Structural Biology X* 4 10020 (2020)

[3] Wrapp et al., “Cryo-EM structure of the 2019-nCoV spike in the prefusion conformation”, *Science* 367(6483) 1260-1263 (2020)

[4] Davis et al., “Modular Assembly of the Bacterial Large Ribosomal Subunit” *Cell* 167(6) 1610-1622 e1615 (2016)

%%

2) Modeling statistics: The authors include modeling statistic comparisons for the TRP channel but not for reconstructions. I think these values should be generated for atomic-resolution maps such as apoferritin and the spike protein so that readers can easily see whether this new software improved modeling statistics. Right now, the reader is left unknown as to whether improvements that the authors saw for the TRP channel extend to other maps (it appears that it will, but having modeling statistics will confirm this). This is particularly important for the new model building performed on the spike protein; a ‘Table 1’ description needs to be provided for modeling information and statistics.

Following the referee recommendation, we have included EMRINGER and MolProbit modeling statistic comparisons for the TRP channel, apoferritin and spike S protein.

The EMRINGER results obtained are:

Apoferritin

EMRinger LocSpiral: 8.63

EMRinger Relion: 2.51

Spike S protein

EMRinger LocSpiral: 2.31

EMRinger Relion: 2.27

We have included the new Table S1 and the following sentences:

1) Table S1

		TRP channel (EMD-10418) (PDB 6t9n)	Apoferritin (EMD-9865) (PDB 6v21)	SARS-CoV-2 (EMD-21375) (PDB 6vsb)
EMRINGER	EMRINGER LocSpiral	2.31	8.63	2.31
	EMRINGER Relion	2.36	2.51	2.27
	Rotamer-ratio LocSpiral	0.70	0.97	0.70
	Rotamer-ratio Relion	0.72	0.67	0.73
	Max Z-score LocSpiral	7.93	48.83	9.47
	Max Z-score Relion	8.11	14.22	9.06
	Model Length LocSpiral	1184	3200	1683
	Model Length Relion	1184	3200	1598
MOLPROBITY	All-atom Clashscore LocSpiral	6.44	5.90	13.66
	All-atom Clashscore Relion	6.12	5.27	14.34
	Ramachandran Plot LocSpiral	Outliers:0.00% Allowed:4.38% Favored:95.62%	Outliers:0.00% Allowed:1.90% Favored:98.10%	Outliers:0.00% Allowed:8.36% Favored:91.46%
	Ramachandran Plot Relion	Outliers:0.00% Allowed:3.12% Favored:96.88%	Outliers:0.00% Allowed:2.33% Favored:97.67%	Outliers:0.00% Allowed:8.54% Favored:91.64%
	Rotamer Outliers LocSpiral	7.24 %	1.39 %	13.63 %
	Rotamer Outliers Relion	2.77 %	1.49 %	9.96 %
	Cbeta Deviations LocSpiral	0.00 %	0.00 %	0.00 %
	Cbeta Deviations Relion	0.00 %	0.00 %	0.00 %
	Peptide Plane LocSpiral	Cis-proline:0% Cis-general:0% Twisted Proline:0% Twisted-General: 0%	Cis-proline:25% Cis-general:0% Twisted Proline:0% Twisted- General:0%	Cis-proline:0.67% Cis-general:0% Twisted Proline:0.67% Twisted- General:0.03%
	Peptide Plane Relion	Cis-proline:0% Cis-general:0% Twisted Proline:0% Twisted-General: 0%	Cis-proline:25% Cis-general:0% Twisted Proline:0% Twisted- General:0%	Cis-proline:0% Cis-general:0% Twisted Proline:0% Twisted-General: 0%

Table S1 EMRINGER and MolProbity modeling scores obtained between sharpened maps by Relion postprocessing and LocSpiral, and corresponding atomic models after refining the structure against corresponding maps by Phenix real_space_refine approach using 5 refining iterations.

2) At the end of the Apoferritin subsection:

“We compute also EMRINGER and MolProbity scores (Barad, Echols et al. 2015) between these maps (EMD-9865 and LocSpiral) and the atomic model (PDB 6v21) after refining the structure against corresponding maps by Phenix real_space_refine approach (Afonine, Poon et al. 2018) using 5 refining iterations. The results obtained are shown in Table S1.”

3) At the end of the SARS-CoV-2 subsection:

“Corresponding EMRINGER and MolProbity scores, calculated between LocSpiral map and the improved atomic model, and between EMD-21375 and the deposited model (PDB 6vsvb), are shown in Table S1. In both cases the atomic structures were refined against corresponding maps by Phenix real_space_refine approach (Afonine, Poon et al. 2018) using 5 refining iterations.

%%%%%%%%%

3) *The authors performed single particle analysis for the spliceosome. As such, for rigor and transparency the authors should provide a description in the methods sections regarding data processing.*

Following the referee recommendation, we have included a new subsection in the methods section to provide a detailed description of the followed spliceosome image processing steps. For the sake of clarity, we reproduce here this new information:

Cryo-EM image processing of the spliceosome data

The dataset is composed of 327,490 particle images of a spliceosomal B-complex from yeast (EMPIAR-10180) (Plaschka et al., 2017). The particles were polished with Relion, downsampled to 1.699 Å/px and windowed to a size of 320x320 pixels. As described in (Gomez-Blanco, Kaur et al. 2019) a set of 30 initial volumes were obtained by RANSAC (15 maps) and Eman2 (15 maps) and processed by volume selector approach producing two different initial volumes. Then, Relion 3D classification was used to compute two classes providing both volumes as reference initial maps (class 1 and class 2 composed by 201,407 and 126,083 particles respectively). The resulting classes were refined by Relion auto-refine using the maps obtained in the previous 3D classification. Finally, Relion postprocessing provided maps at 4.28 Å and 4.58 Å for class 1 and class 2, respectively. Lastly, local resolution was calculated using Relion for both classes.

%%%%%%%%%

REVIEWERS' COMMENTS

Reviewer #2 (Remarks to the Author):

The authors have clearly addressed my questions. I believe in the additional text and reporting has helped to provide a more clear manuscript.